# Application of Gas-Liquid Microextraction (GLME)/GC-MS for Flavour and Fragrance in Ice Cream Detection and Composition Analysis

**DOI:** 10.3390/molecules28020522

**Published:** 2023-01-05

**Authors:** Li Mu, Qi Tong, Yuhang Liu, Xianglong Meng, Peng He, Gang Li, Linyang Ye

**Affiliations:** 1College of Food Science and Engineering, Changchun University, Changchun 130022, China; 2Jilin Province Product Quality Supervision and Inspection Institute, Changchun 130023, China; 3Institute of Vegetables and Flowers, Chinese Academy of Agricultural Sciences, Beijing 100081, China

**Keywords:** ice cream, gas-liquid microextraction (GLME), food composition analysis, classification of functional groups, principal component analysis

## Abstract

Under gas-liquid microextraction (GLME) operating conditions (extraction temperature 270 °C, extraction time 7 min, condensation temperature −2 °C, and carrier nitrogen gas speed 2.5 mL/min), ice cream samples, as a representative food, were pre-treated. The volatile aroma components of each sample was qualitatively analysed using GC-MS. The principal component analysis was conducted to classify the functional groups, which showed that alcohols, acids, esters, ketones, and aldehydes were the main compounds responsible for the aroma of ice cream. It was found that furan-3-carboxaldehyde, 3-furanmethanol, 2(5*H*)-furanone, 5-methylfuranal, 2,5-diformylfuran, 3-hydroxy-2-methyl-4-pyrone, 5-hydroxymethylfurfural, ethyl maltol, and glycerol were routinely used flavour ingredients in ice cream.

## 1. Introduction

With the rapid development of modern society, people are no longer just looking for satiety, health, and nutrition, but also have higher requirements for the flavour of food. The use of flavourings in food products can improve the diversity of food flavours, while aroma components are used as an index to judge the use of flavourings in foods. Due to the complex and unstable composition of volatile substances responsible for flavour, oxidation, polymerization, and condensation, reactions may occur during the extraction of volatile substances.

Flavours and fragrances are usually attached to complex food matrices, which also affect the composition analysis of flavours and fragrances [1,2]. Steam distillation [3,4], simultaneous distillation [5,6], static-headspace distillation [7,8,9], vacuum distillation [10], supercritical CO_2_ extraction [11,12], and solid-phase microextraction [13,14] techniques were widely used to extract volatile substances [15]. Each extraction method has its own advantages and disadvantages. Yang C et al. [16] used optimised gas-liquid microextraction parameters and detected alkylphenols in seafood by liquid chromatography. Nan et al. [17] used gas-liquid microextraction and gas chromatography-mass spectrometry to create a method for the rapid detection of multiple organophosphorus pesticide residues in edible mushrooms. Maltol and ethyl maltol are often added as food flavour enhancers in baked goods, ice cream, and maltol candy. In this experiment, gas-liquid microextraction (GLME)/GC-MS for food flavour and fragrance detection and composition analysis. The method is simple, quick, and reproducible, and only requires several milligrams for each analysis, which takes only a few minutes. Ethyl maltol as representative substance is used as general additive in ice cream, and the gas-liquid microextraction GLME /GC-MS detection method for flavour and fragrance is established.

Compared with other conventional additives, there are few systematic studies on the flavours and fragrances in food. The flavours and fragrances mentioned in China GB 2760-2014 and encoded in the Flavor and Extract Manufacturers Association (FEMA) of the United States are used as the principal component analysis and research, the principal component analysis of ice cream with a high use rate of flavours and fragrances were conducted through classification of functional groups.

## 2. Results and Discussion

### 2.1. Optimization of GLME Operating Conditions

#### 2.1.1. Extraction Temperature

The ethyl maltol standard solution was pre-treated via GLME with extraction time of 7 min, carrier gas (Nitrogen) speed of 2.5 mL/min, and condensation temperature of −2 °C conditions were unchanged, samples were extracted at 240, 250, 260, 270, 280, and 290 °C, respectively. The detection progress was monitored according to the GLME working principle and operation conditions in Section 3.6. Through the GC-MS analysis, the influence of different extraction temperatures on the spiked recovery of ethyl maltol was investigated. As shown in Figure 1a, the spiked recovery rate was shown an increasing trend when the extraction temperature was within the gradient of 240 °C–270 °C, while the spiked recovery rate of the ethyl maltol standards were shown a decreasing trend when the extraction temperature was within the gradient of 270 °C–290 °C. This phenomenon of increasing temperature and decreasing spiked recovery rate is related to the structural changes of the ethyl maltol standards, which may have been accompanied by cleavage reactions or formation of isomers while the temperature is increasing, thus the optimal extraction temperature should be around 270 °C.

#### 2.1.2. Extraction Time

Ethyl maltol standards were extracted at 5, 6, 7, 8, and 9 min, respectively, according to the GLME working principle and operation conditions in Section 3.6. Through GC-MS analysis, the influence of different extraction time on the spiked recovery rate of ethyl maltol samples were investigated. As shown in Figure 1b, the spiked recovery rate of ethyl maltol standard was shown that a slow increase in the extraction time from 5 min to 7 min; after 7 min, the spiked recovery rate tended to level off. It was indicated that when the extraction time was used as a single variable, after the extraction time reached a certain threshold value, too much extraction time would not have a great impact on the experiment. Thus, the optimal extraction time should be around 7 min.

#### 2.1.3. Carrier Gas Speed

According to the GLME working principle and operation conditions in Section 3.6. Ethyl maltol standards were extracted by carrier gas speed at 1.0 mL/min, 1.5 mL/min, 2.0 mL/min, 2.5 mL/min, and 3.0 mL/min, respectively. As shown in Figure 1c, within practice operation, the spiked recovery rate of ethyl maltol sample was significantly improved when the carrier gas speed was in the gradient range of 1.0–2.5 mL/min. When the gas flow rate exceeds over 2.5 mL/min, the spiked recovery rate of ethyl maltol sample was decreases. It has been demonstrated that, although a fast gas flow rate can theoretically increase the contact area, too fast a gas flow rate in a confined space makes the purge state unstable and prevents the volatile components in the sample from being trapped by the purge in a stable manner.

#### 2.1.4. Condensation Temperature

The condensation temperature was changed at −5 °C, −4 °C, −3 °C, −2 °C, −1 °C, and 0 °C, respectively, according to the GLME working principle and operation conditions in Section 3.6. The effect of different condensation temperatures on the spiked recovery rate of the ethyl maltol samples were investigated by GC-MS analysis, as shown in Figure 1d. It can be known that the influence of condensation temperature on the extraction progress of GLME was more significant. When the condensation temperature were increased from −5 °C to −2 °C, the spiked recovery rate of the samples tended to increase obviously, and when the condensation temperature was at −2 °C, the spiked recovery rate was at the highest value, when the condensation temperature continued to increase, the spiked recovery rate was decreased, which indicated that the condensation temperature needs to reach a certain level in order to complete the sample transformation from gas phase to liquid phase in order for the volatile substances extracted by GLME to be received back as much as possible; thus −2 °C was chosen as the optimal condensation temperature in this study.

#### 2.1.5. Response Surface Design Optimization

Design Experts 12 was used to optimize the design by response surface design, and the spiked recovery rate of ethyl maltol sample was used as the response variable. The four factors, extraction temperature (A), extraction time (B), carrier gas speed (C), and condensation temperature (D), had a large effect on the spiked recovery rate of ethyl maltol, and the experimental data were designed and processed.

In order to find the optimal GLME extraction conditions for the ethyl maltol samples, a binary multiple regression equation was constructed for the four factors, extraction temperature (A), extraction time (B), carrier gas speed (C), and condensation temperature (D), with the spiked recovery rate (Y) of ethyl maltol samples, as shown in the following Equation (1):(1)$$\begin{array}{l} Y=78.90+0.8575A+0.1525B-0.3475C+0.1192D+0.0550AB\\ -0.2700AC-0.0325AD+0.1025BC+0.3000BD+0.4100CD\\ -1.11A^{2}-0.2035B^{2}-0.3660C^{2}-0.2185D^{2} \end{array}$$

In the regression equation fitted by the response surface model, a positive coefficient of the linear phase indicates that the factor acts synergistically on the response value, a positive correlation between the response value and the factor. Conversely, a linear phase with a negative coefficient indicates an antagonistic effect on the response value, a negative correlation between the response value and the factor. From the above equation, it was clearly seen that A, B, C, D, AC, BC, BD, and CD acted synergistically on the spiked recovery rate of ethyl maltol samples.

The analysis of variance (ANOVA) in Design Expert 12 is mainly used to verify the significance and generalizability of the regression equation for the experimental data (Table 1). A larger F-value proves that the fit is more significant, and, when the *p*-value is less than 0.05, it can prove the goodness of fit of the model. The interaction factors with *p*-values less than 0.05 have AC, BD and CD (Figure 2), when the stronger the interaction between the two factors, the more significant the effect on the response value, the more curved and steep the fitted response surface is, the more dense the contours are, and the more elliptical the shape tends to be. The response value R^2^ was 0.9898, which proves that the model of response surface simulation was more reliable, and the predicted accuracy of each parameter of GLME operating conditions was more reliable.

The optimal GLME operating conditions were optimized and obtained according to the actual situation, namely: extraction temperature of 270 °C, extraction time of 7 min, gas flow rate of 2.5 mL/min, and condensation temperature of −2 °C. The average yield of response value ethyl maltol spiked recovery rate was 78.90%, which was consistent with the optimized result of response value of 78.90%.

### 2.2. Principal Components(Pcs) Analysis of Ice Cream

#### 2.2.1. GC-MS Analysis

Ten ice cream samples were analysed by GC-MS to contain components with ≥80% similarity (Table 2), and the chromatogram of samples were obtained. In Figure 3, a total of 85 volatile components were detected in 10 ice cream samples. The identified components were grouped according to functional groups and the classification results were as follows: 8 alcohols, 13 acids, 22 esters, 8 aldehydes, 15 ketones, 3 phenols, 11 heterocycles, 3 alkanes, 2 amines, and 1 ether. Among the identified components, esters accounted for the largest proportion (25.58%), followed by ketones (17.44%), and acids (15.12%). Ether accounted for the smallest proportion (1.16%) (Figure 4a).

#### 2.2.2. Analysis of Volatile Aroma Components

Among the 85 volatile components identified in the 10 ice cream samples, 13 co-occurring components were detected, namely formic acid, acetic acid, glycerol, 3-furanomethanol, acetone alcohol, furan-3-carboxaldehyde, 5-methylfurfural, 5-hydroxymethylfurfural, 5-acetoxymethyl-2-furaldehyde, methyl maltol, methylcyclopentenolone, 2(5*H*)-furanone, and methyl furfurate. Among them, acetic acid and glycerine are generally used as flavourings and fragrances in food to add flavour to ice cream, and are also common ingredients in food processing. Additionally, 3-Furanmethanol is not codified by FEMA; however, according to the literature review, this substance has the aroma of roasted potatoes [18]. Additionally, 3-Furfural has a sweet aroma similar to fermented bread with charred aroma [19]. In addition, 5-Methyl furfural has a caramel rich sweet aroma [20], and the source of this aroma is mainly the Maillard reaction [21], 5-hydroxymethyl furfural has a sweet characteristic flavour similar to caramel or coffee, 5-acetoxymethyl-2-furaldehyde has a floral aroma [22], methyl maltol embodies a creamy caramel sweet aroma, methyl cyclopentenol ketone will embody the aroma of liquorice after dilution, which is allowed to be used in foods, such as creamy hard candy in China’s GB 2760-2014; 2(5*H*)-furanone has a fruity sweet aroma [23]. When used as a synthetic flavouring, methyl furoate will embody different aromas at different dilutions, possessing the flavour of caramel at low dilutions, and releasing a fruity aroma at high dilutions [24].

In addition, among the 10 samples studied, some components, such as 4-hydroxy-2,5-dimethyl-3(2*H*)furanone, (+)-limonene, and pineapple ester, were also detected in individual ice cream samples. Upon review, 4-hydroxy-2,5-dimethyl-3(2*H*)furanone can impart baking and caramel aromas to foods, (+)-limonene has citrus and lemon aromas, which was associated with the orange flavour of the samples, pineapple ester, as implied by the name, can add pineapple aroma to foods [25]; therefore, this component was detected in the pineapple flavoured samples.

Statistical analysis of principal components(PCs) was performed on the processed data by SPSS 23.0 software.

In principal component analysis, factor analysis can simplify complex multivariate factors by using as few factors as possible to analyse the influence of volatile aroma components in samples (Figure 4b and Figure 5). Before the factor analysis of volatile aroma components in ice cream samples, the content of phenols, alkanes, ethers, and amines in all components was low, therefore, the above four substances were removed from the factor analysis before analysis was performed. The characteristic roots and contribution rate (percent variance) of the principal components (PCs) in the ice cream sample were used as the basis for selecting the principal components (PCs). It can be observed that the contribution of characteristic roots greater than 1 come from the first three factors. Through the contribution rate was shown that the total variance of the first three factors reached 78.170%. That is, a three-factor model explained 78.170% of the entire analysed data.

The functional groups with high positive correlation in the first principal factor (PC1) were aldehydes and heterocycles in Figure 4b, while alcohols showed high negative correlation, esters and acids showed high positive correlation in the second principal factor(PC2), and ketones showed high positive correlation in the third principal factor(PC3). It can be further inferred from the total variance contribution of the first two principal factors above 50% in Figure 4. that aldehydes, esters, alcohols, acids, ketones, and heterocyclic groups were the characteristic functional group categories in the volatile aroma of ice cream.

### 2.3. Quantitative Analysis of Samples

According to Section 3.8, the quantitative results of functional groups detected in randomly selected ice cream were calculated. Statistical analysis was conducted on functional groups categories and concentrations of each sample, as shown in Figure 6.

It could be seen that the quantities of aldehydes were the largest in all detected functional groups. All samples contained aldehydes, such as furfural, furan-3-carboxaldehyde, 5-methylfurfural and 5-hydroxymethylfurfural, which give them a sweet caramel flavour. Esters mostly contribute to a fruity fragrance [26,27].

## 3. Materials and Methods

### 3.1. Experimental Materials and Reagents

Different flavours and manufacturers of ice cream were purchased from a retail market.

Methanol (chromatographic purity, 95%) was purchased from TEDIA Co., Ltd (Tedia Way Fairfield, OH, USA). Anhydrous sodium sulphate (analytical purity) was purchased from Sinopharm Chemical Reagent Co., Ltd (Shanghai, China). It was heated at 400 °C for 12 h, then cooled, and stored in a dryer before use. Quartz wool (Aoreilong New Materials Technology Co., Ltd. Shandong, China) was activated at 400 °C for 10 h before use. A C_19_ normal alkane standard solution (chromatographic purity, 99.9%) was purchased from Aladdin (Shanghai, China). Dichloromethane (chromatographic purity, 95%) was purchased from TEDIA Co., Ltd. (Tedia Way Fairfield, OH, USA) Ethyl maltol (purity, 99%)was purchased from J&K Scientific Co., Ltd. (Beijing, China).

### 3.2. Experimental Instruments

Gas-liquid microextraction (ME-101) developed by the Key Laboratory of Changbai Mountain Biological Resources and Functional Molecules, the Ministry of Education, Yanbian University (Figure 7). Gas chromatography-mass spectrometry (GC-MS-QP2010), Shimadzu (Tokyo, Japan) was used.

### 3.3. GC-MS Conditions

#### 3.3.1. Gas Chromatography Conditions

GC was performed using a 0.25 µm, 30 m × 0.25 mm J and W DB-5 quartz capillary column. The heating procedure involved holding at 40 °C for 2 min, and increasing to 150 °C at 5 °C/min, then heating to 280 °C at 15 °C/min and holding for 5 min.

#### 3.3.2. Mass Spectroscopy Conditions

The carrier gas was helium (He), purity ≥99.999%, with a flow rate of 1.78 mL/min. The injection volume was 1.0 µL. The injection mode was non-shunt injection. The ionization mode was electron bombardment (EI). The impact power was 70 eV. The GC-MS interface temperature was 300 °C. The ion source temperature was 230 °C. The full-scan mode was used, with a mass scan range of *m*/*z* 29–500, and a solvent delay of 2.5 min.

### 3.4. Sample Preparation

The ice cream were homogenized as samples by homogenizer, and numbered ice cream separately from 01 to 10; these samples were stored at −4 °C.

### 3.5. Preparation of Standard Solution

Ethyl maltol reserve solution was prepared by precisely weighing 0.1 g ethyl maltol standards, which was then dissolved in methanol and constant volume in a 100 mL brown volumetric flask, to prepared 1000 mg/L standard solution.

C_19_ n-alkanes (0.0100 g) were accurately weighed in a 100 mL volumetric flask and dissolved in a fixed volume of dichloromethane to prepare 100.0 mg/L of the internal standard solution.

### 3.6. Glme Working Principle and Operation Conditions

A 10 µL ethyl maltol standard solution or a 0.100 g sample was precisely weighed into a sample tube filled with quartz glass wool and an appropriate amount of anhydrous sodium sulphate. Then, 10 µL of the internal standard solution was added to the sample tube. Afterwards, the sample tube was moved to the sample pool, and the sample pool was sealed with a rubber stopper. An amount of 50 µL methanol extraction phase was added to a 200 µL inner lining tube as the accepting phase and placed in the condensation tank. GLME conditions were set as: extraction temperature of 270 °C, carrier gas (Nitrogen) speed of 2.5 mL/min, condensation temperature of −2 °C, and extraction time of 7 min. After running the GLME program to the termination of the set time, the acceptor phase volume was eluted and fixed to 100 µL, and the sample was mixed by vortex shaking and finally examined by GC-MS analysis [28].

### 3.7. Principal Component Analysis

This study aims to use the idea of dimensionality reduction to transform multiple indicators into a few comprehensive indicators (i.e., principal components), where each principal component can reflect most of the information of the original variables, and the contained information does not repeat.

### 3.8. Quantitative Calculation of Samples

Calibration was performed using C_19_ as an internal standard to quantify the components containing standards. Assuming that the relative response factor was 1 and the recovery rate was 100%, the provisional determination of components without the standard was quantified relative to the internal standard. The specific formula used to calculate the compounds’ mass (Mi) from the peak area ratio is shown in Equation (2), and the specific formula used to calculate the concentration of substances is shown in Equation (3):M_i_ = A_i_/A_s_ × F_i_ × M_s_(2)
C_i_ = M_i_/m(3)

In Equations (2) and (3), $A_{i}$ represents the peak area of the designated component, $A_{s}$ is the peak area of the internal standard, $M_{i}$ is the mass of the designated component, $M_{s}$ is the mass of the internal standard, $F_{i}$ is the relative correction factor, which is assumed to be 1, $C_{i}$ is the concentration of the designated component, *m* is the mass of sample.

## 4. Conclusions

Qualitative and quantitative analyses of volatile flavours and fragrances in ice cream were conducted by a gas-liquid microextraction/GC-MS method. Statistically, it was found that alcohols, acids, esters, ketones, aldehydes, and heterocycles were the principal components in the aromas of ice cream samples. Additionally, Furan-3-carboxaldehyde, 3-furanmethanol, 2(5*H*)-furanone, 5-methylfurfural, 2,5-diformylfuran, 3-hydroxy-2-methyl-4-pyrone, 5-hydroxymethylfurfural, ethyl maltol, and glycerine were routinely used flavour components. The applicability and safety contents of these components can be obtained in China GB 2760-2014 and in the Flavor and Extract Manufacturers Association (FEMA) of the United States. We summarized the conventional flavours and fragrances used in casual foods, represented by ice cream, to provide an effective reference basis for future research on flavour and fragrance components in foods.

With the improvement of people’s living standards, the variety and taste of food is increasing. Thus, the use of flavours and fragrances will also increase. In future research work, it is possible to combine computer technology, statistics, and other methods to build a library of characteristic fingerprint profiles of flavours and fragrances in the context of a large amount of data, enabling the analysis of flavours and fragrances to reach a state of cloud sharing.

## Figures and Tables

**Figure 1 molecules-28-00522-f001:**
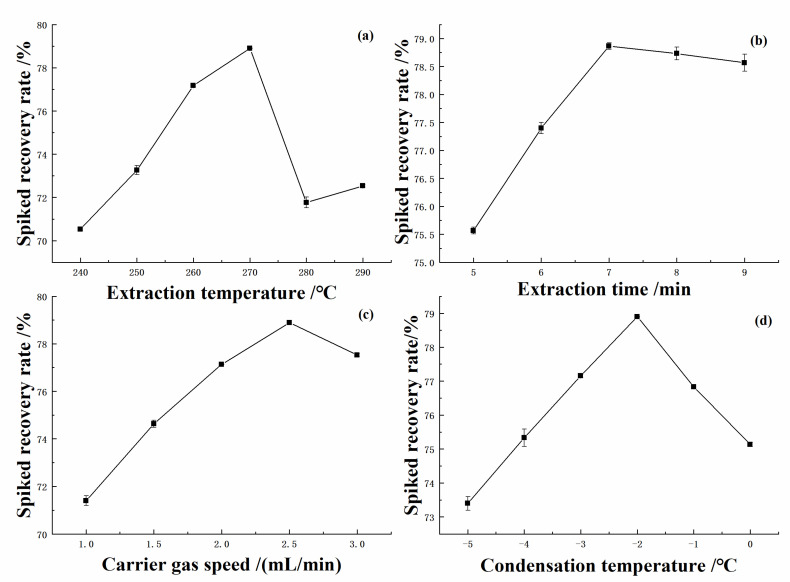
Effect of four factors to spiked recovery rate: extraction temperatures (**a**), extraction time (**b**), carrier gas speed (**c**), and condensation temperature (**d**).

**Figure 2 molecules-28-00522-f002:**
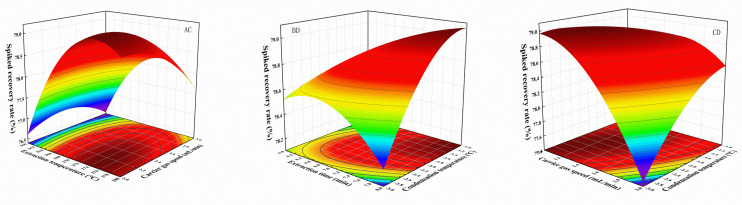
Effect of factors AC, BD, and CD interaction response surface.

**Figure 3 molecules-28-00522-f003:**
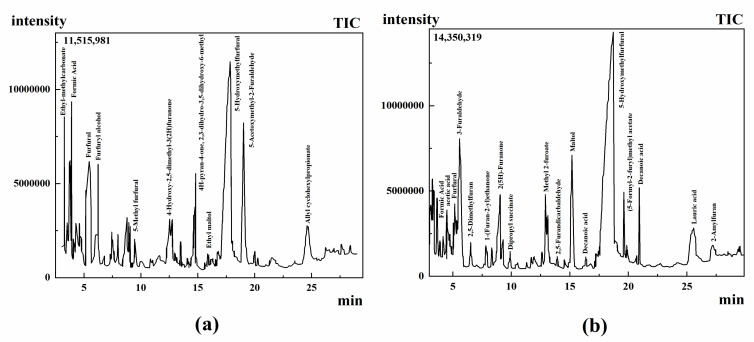
Chromatograms of ice cream sample 1 (**a**) and sample 2 (**b**).

**Figure 4 molecules-28-00522-f004:**
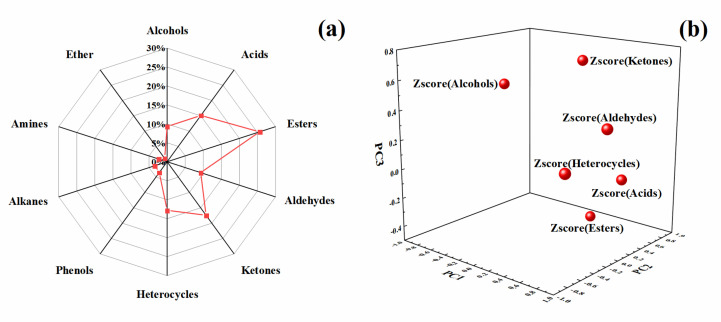
Ratio of functional groups of ice cream samples (**a**) and Volatile aroma loading diagram of ice cream samples (**b**).

**Figure 5 molecules-28-00522-f005:**
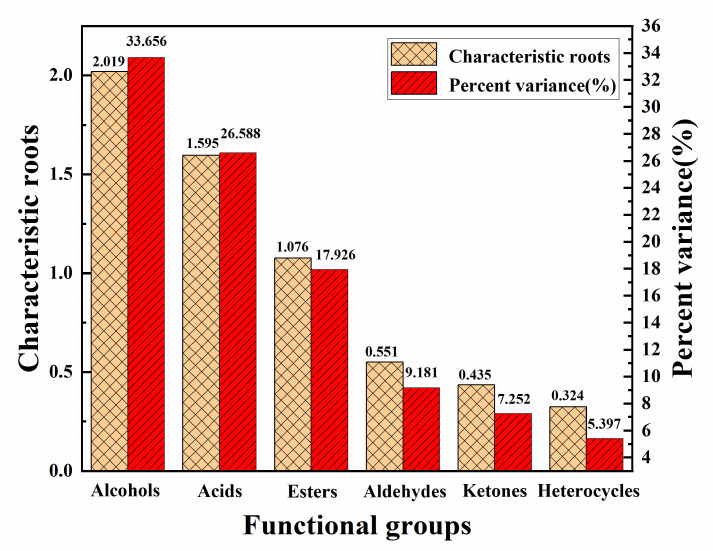
Results of variance analysis of volatile aroma components in ice cream samples.

**Figure 6 molecules-28-00522-f006:**
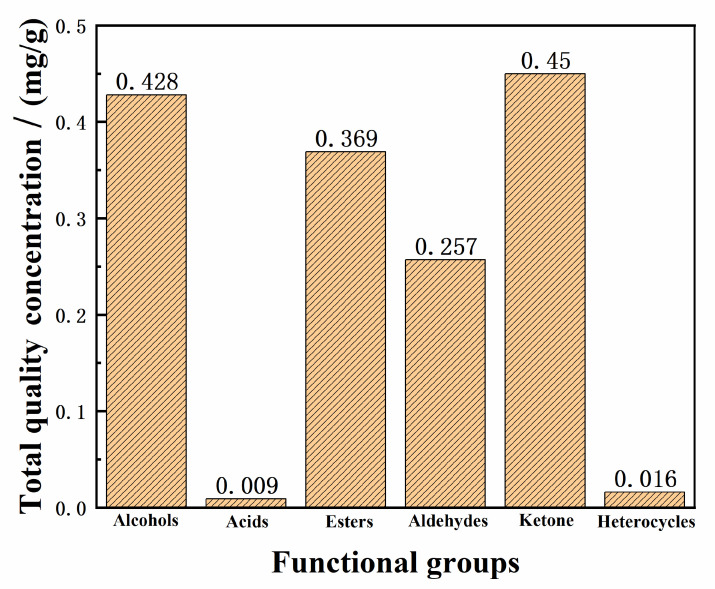
Total quality concentrations of the functional groups in ice cream samples.

**Figure 7 molecules-28-00522-f007:**
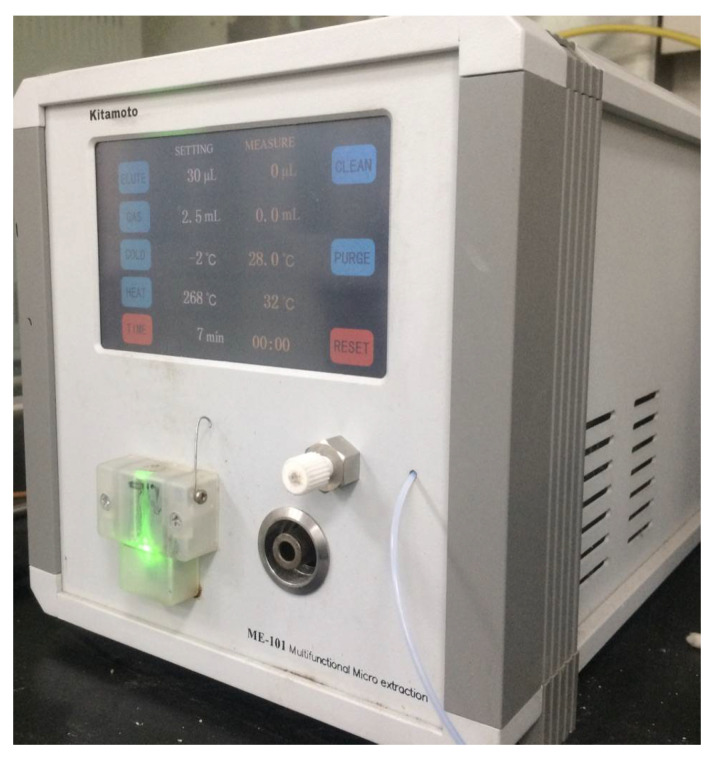
Gas-liquid microextraction.

**Table 1 molecules-28-00522-t001:** Response to ANOVA for the spiked recovery rate of ethyl maltol samples.

Scheme	Sum of Squares	df	Mean SQUARE	F-Value	*p*-Value	
Model	20.28	14	1.45	97.37	<0.0001	significant
A-Extraction temperature	8.82	1	8.82	593.04	<0.0001	
B-Extraction time	0.2791	1	0.2791	18.76	0.0007	
C-Carrier gas speed	1.45	1	1.45	97.39	<0.0001	
D-Condensation temperature	0.1704	1	0.1704	11.45	0.0045	
AB	0.0121	1	0.0121	0.8132	0.3824	
AC	0.2916	1	0.2916	19.60	0.0006	
AD	0.0042	1	0.0042	0.2840	0.6025	
BC	0.0420	1	0.0420	2.82	0.1150	
BD	0.3600	1	0.3600	24.20	0.0002	
CD	0.6724	1	0.6724	45.19	<0.0001	
A²	8.01	1	8.01	538.13	<0.0001	
B²	0.2687	1	0.2687	18.06	0.0008	
C²	0.8690	1	0.8690	58.40	<0.0001	
D²	0.3097	1	0.3097	20.82	0.0004	
Residual	0.2083	14	0.0149			
Lack of Fit	0.2083	10	0.0208	1.041 × 10^5^	<0.0001	significant
Pure Error	8.000 × 10^−7^	4	2.000 × 10^−7^			
Cor Total	20.49	28				

**Table 2 molecules-28-00522-t002:** Identification results of ice cream samples 1–10 by GC-MS.

Sample		1		Sample		2	
Retention time/min	CAS	Compound	FEMA	Retention time/min	CAS	Compound	FEMA
3.395	623-53-0	Ethyl-methylcarbonate	-	3.55	64-18-6	Formic Acid	2487
3.55	64-18-6	Formic Acid	2487	3.825	64-19-7	Acetic acid	2006
3.68	56-81-5	Glycerol	2525	3.995	116-09-6	Acetol	4462
3.850	64-19-7	Acetic acid	2006	4.6	56-81-5	Glycerol	2525
3.99	57-55-6	Propylene Glycol	2940	4.935	57-55-6	Propylene Glycol	2940
4.1	116-09-6	Acetol	4462	5.35	625-86-5	2,5-Dimethylfuran	4106
4.59	68-12-2	*N*, *N*-Dimethylformamide	-	5.75	498-60-2	3-Furaldehyde	3737
4.975	107-92-6	Butyric Acid	2221	6.8	4412-91-3	3-Furancarbinol	-
5.485	98-01-1	Furfural	2489	7	1759-71-3	cis-1,2-Cyclohexanediol diacetate	-
5.58	498-60-2	Furan-3-carboxaldehyde	3737	7.775	1192-62-7	1-(Furan-2-yl)ethanone	3163
6.085	4412-91-3	3-Furanmethanol	-	8.1	497-23-4	2(5*H*)-Furanone	4138
6.24	98-00-0	Furfuryl alcohol	2491	9.305	620-02-0	5-Methyl furfural	2702
6.86	606-45-1	Methyl 2-methoxybenzoate	2717	10.02	924-88-9	Diisopropyl succinate	-
7.195	108-94-1	Cyclohexanone	3909	11.45	80-71-7	Methyl cyclopentenolone	2700
7.64	1192-62-7	2-Acetylfuran	3163	13.135	611-13-2	Methyl 2-furoate	2703
8.4	591-11-7	BETA-ANGELICA LACTONE	4438	14.14	118-71-8	Maltol	2656
8.96	16867-04-2	2,3-Dihydroxypyridine	-	16.535	124-07-2	Octanoic acid	2799
9.175	620-02-0	5-Methyl furfural	2702	17.6	4412-96-8	3-Methyl-2-furoic acid	-
9.8	925-15-5	Dipropyl succinate	-	19.06	67-47-0	5-Hydroxymethylfurfural	-
11.84	110-13-4	Acetonyl acetone	-	19.97	10551-58-3	(5-Formyl-2-furyl)methyl acetate	-
12.485	3658-77-3	4-Hydroxy-2,5-dimethyl-3(2*H*)furanone	3174	20.625	4282-34-2	2,5-Thiophenedicarboxylic acid dimethyl ester	-
12.835	504-15-4	Orcinol	3102	21.86	334-48-5	Decanoic acid	2364
13.07	696-11-7	1-methyl-1,3-diazinane-2,4-dione	-	24.275	706-14-9	γ-Decanolactone	2360
15.025	28564-83-2	4*H*-pyran-4-one, 2,3-dihydro-3,5-dihydroxy-6-methyl-	-	26.275	143-07-7	Lauric acid	2614
16.245	1073-96-7	3,5-dihydroxy-2-methylpyran-4-one	-	27.1	4437-22-3	2,2’-(Oxybis(methylene)) difuran	3337
16.49	4940-11-8	Ethyl maltol	3487	27.395	77-93-0	Triethyl citrate	3083
17.845	67-47-0	5-Hydroxymethylfurfural	-	28.635	544-63-8	Myristic acid	2764
19.83	10551-58-3	5-Acetoxymethyl-2-Furaldehyde	-				
23.145	2705-87-5	Allyl cyclohexylpropionate	2026				
**Sample**		**3**		**Sample**		**4**	
**Retention time/min**	**CAS**	**Compound**	**FEMA**	**Retention time/min**	**CAS**	**Compound**	**FEMA**
3.57	64-18-6	Formic Acid	2487	3.415	64-18-6	Formic Acid	2487
3.855	64-19-7	acetic acid	2006	3.71	64-19-7	acetic acid	2006
4.1	116-09-6	acetol	4462	3.92	116-09-6	acetol	4462
4.61	56-81-5	glycerol	2525	4.46	56-81-5	glycerol	2525
5.13	617-35-6	Ethylpyruvate	2457	5.315	98-01-1	Furfural	2489
5.35	2041-15-8	1,3,5-Cyclohexanetriol	-	5.77	498-60-2	3-Furaldehyde	3737
5.75	498-60-2	3-Furaldehyde	3737	6.7	4412-91-3	3-Furancarbinol	-
6.75	4412-91-3	3-Furancarbinol	-	7.75	1192-62-7	1-(Furan-2-yl)ethanone	3163
8.05	497-23-4	2(5*H*)-Furanone	4138	8.01	497-23-4	2(5*H*)-Furanone	4138
8.96	16867-04-2	hydroxypyridone	-	8.985	2361-27-5	2-Thiophenecarbohydrazide	-
9.255	620-02-0	5-Methyl furfural	2702	9.285	620-02-0	5-Methyl furfural	2702
10.025	924-88-9	Diisopropyl succinate	-	10	925-15-5	Dipropyl succinate	-
10.46	765-70-8	Maple lactone	-	10.475	637-88-7	1,4-Cyclohexanedione	-
11.235	5989-27-5	(+)-Limonene	2633	11.445	80-71-7	Methyl cyclopentenolone	2700
11.405	80-71-7	Methyl cyclopentenolone	2700	11.975	110-13-4	Acetonylacetone	-
11.985	3128-07-2	6-Oxoheptanoic acid	-	12.96	823-82-5	2,5-Furandicarbaldehyde	-
12.46	1192-62-7	1-(Furan-2-yl)ethanone	3163	13.115	611-13-2	Methyl 2-furoate	2703
12.975	3658-77-3	4-Hydroxy-2,5-dimethylfuran-3(2*H*)-one	3174	14.11	118-71-8	Maltol	2656
13.1	611-13-2	Methyl 2-furoate	2703	16.19	1193-79-9	1-(5-Methylfuran-2-yl)ethanone	3609
13.635	590-86-3	Isovaleraldehyde	2692	19.225	67-47-0	5-hydroxymethylfurfural	-
13.92	98-00-0	Furfuryl alcohol	2491	20	10551-58-3	(5-Formyl-2-furyl)methyl acetate	-
14.055	118-71-8	Maltol	2656	20.99	102-76-1	Triacetin	2007
17.4	18720-62-2	2-methylheptan-3-ol	-	25.855	498-07-7	Levoglucosan	-
17.56	10551-58-3	(5-Formyl-2-furyl)methyl acetate	-	28.605	544-63-8	Myristic acid	2764
18.85	67-47-0	5-hydroxymethylfurfural	-	29.52	84-69-5	Diisobutyl phthalate	-
21.76	112-37-8	Undecanoic acid	3245				
23.16	2705-87-5	Allyl 3-cyclohexylpropionate	2026				
27.09	3777-69-3	2-Amylfuran	3317				
28.605	544-63-8	Myristic acid	2764				
**Sample**		**5**		**Sample**		**6**	
**Retention time/min**	**CAS**	**Compound**	**FEMA**	**Retention time/min**	**CAS**	**Compound**	**FEMA**
3.155	64-18-6	Formic Acid	2487	3.055	64-18-6	Formic Acid	2487
3.43	64-19-7	acetic acid	2006	3.33	116-09-6	acetol	4462
3.635	116-09-6	acetol	4462	3.615	627-03-2	Ethoxyacetic acid	-
3.94	79-09-4	Propionic acid	-	3.94	56-81-5	glycerol	2525
4.19	3393-64-4	4-hydroxy-3-methylbutan-2-one	-	4.22	3121-61-7	2-Methoxyethyl acrylate	-
4.245	56-81-5	glycerol	2525	4.475	1117-97-1	*N*-methoxymethanamine	-
5.31	98-01-1	Furfural	2489	5.385	498-60-2	3-Furaldehyde	3737
5.71	498-60-2	3-Furaldehyde	3737	5.65	98-01-1	Furfural	2489
6.555	4412-91-3	3-Furancarbinol	-	6.375	4412-91-3	3-Furancarbinol	-
7.73	1192-62-7	1-(Furan-2-yl)ethanone	3163	6.93	930-60-9	2-Cyclopentene-1,4-dione	-
7.89	497-23-4	2(5*H*)-Furanone	4138	7.73	497-23-4	2(5*H*)-Furanone	4138
8.995	16867-04-2	hydroxypyridone	-	8.85	5380-42-7	2-(Carbomethoxy)thiophene	-
9.27	620-02-0	5-Methyl furfural	2702	8.975	16867-04-2	hydroxypyridone	-
9.81	53119-25-8	1-thiophen-2-ylpentan-1-one	-	9.235	620-02-0	5-Methyl furfural	2702
9.95	925-15-5	Dipropyl succinate	-	9.6	675-10-5	triacetate lactone	-
11.35	80-71-7	Methyl cyclopentenolone	2700	9.75	88-15-3	2-Acetylthiophene	-
11.945	1117-31-3	1,3-butanediol diacetate	-	11.275	80-71-7	Methyl cyclopentenolone	2656
12.965	504-15-4	Orcinol	-	11.885	110-13-4	Acetonylacetone	-
13.07	611-13-2	Methyl 2-furoate	2703	12.035	19432-69-0	Methyl 5-methylthiophene-2-carboxylate	-
13.9	98-00-0	Furfuryl alcohol	2491	12.755	3658-77-3	4-Hydroxy-2,5-dimethylfuran-3(2*H*)-one	3174
14.11	118-71-8	Maltol	2656	12.855	823-82-5	2,5-Furandicarbaldehyde	-
14.65	7492-38-8	2-methyloctan-4-one	-	12.985	611-13-2	Methyl 2-furoate	2703
15.32	1540-29-0	Ethyl 2-acetylhexanoate	4452	13.44	110-62-3	Valeraldehyde	3098
18.9	67-47-0	5-hydroxymethylfurfural	-	13.935	118-71-8	Maltol	2656
19.36	6434-78-2	trans-2-nonene	-	14.625	629-62-9	pentadecane	-
19.96	10551-58-3	(5-Formyl-2-furyl)methyl acetate	-	16.395	501-30-4	kojic acid	-
20.635	4282-34-2	2,5-Thiophenedicarboxylic acid dimethyl este	-	17.505	10551-58-3	(5-Formyl-2-furyl)methyl acetate	-
26.215	498-07-7	Levoglucosan	-	18.38	67-47-0	5-hydroxymethylfurfural	-
27.63	6968-62-3	D-ERYTHRO-l-TALO-OCTONIC ACID, γ-LACTONE	-	20.6	4282-34-2	2,5-Thiophenedicarboxylic acid dimethyl este	-
29.525	84-69-5	Diisobutyl phthalate	-	20.905	102-76-1	Triacetin	2007
				25.5	498-07-7	Levoglucosan	-
				25.89	2311-46-8	propan-2-yl hexanoate	2950
				26.15	143-07-7	Lauric acid	2614
				28.595	544-63-8	Myristic acid	2764
				29.52	84-69-5	Diisobutyl phthalate	-
**Sample**	**7**			**Sample**	**8**		
**Retention time/min**	**CAS**	**Compound**	**FEMA**	**Retention time/min**	**CAS**	**Compound**	**FEMA**
3.25	64-18-6	Formic Acid	2487	3.305	64-18-6	Formic Acid	2487
3.38	141-46-8	Glycolaldehyde	-	3.52	64-19-7	acetic acid	2006
3.46	64-19-7	acetic acid	2006	3.81	116-09-6	acetol	4462
3.68	116-09-6	acetol	4462	4.04	623-53-0	Ethyl-methylcarbonat	-
3.945	623-53-0	Ethyl-methylcarbonat	-	4.305	56-81-5	glycerol	2525
4.235	56-81-5	glycerol	2525	4.6	4254-15-3	(*S*)-(+)-1,2-Propanediol	-
4.565	4254-15-3	(*S*)-(+)-1,2-Propanediol	-	5.3	98-01-1	Furfural	2489
5.3	98-01-1	Furfural	2489	5.71	498-60-2	3-Furaldehyde	3737
5.675	498-60-2	3-Furaldehyde	3737	6.6	4412-91-3	3-Furancarbinol	-
6.575	4412-91-3	3-Furancarbinol	-	7.915	497-23-4	2(5*H*)-Furanone	4138
7.885	497-23-4	2(5*H*)-Furanone	4138	8.995	16867-04-2	hydroxypyridone	-
8.85	22913-26-4	Methyl thiophene-3-carboxylate	-	9.26	620-02-0	5-Methyl furfural	2702
8.985	16867-04-2	hydroxypyridone	-	9.965	924-88-9	Diisopropyl succinate	-
9.235	620-02-0	5-Methyl furfural	2702	11.325	80-71-7	Methyl cyclopentenolone	2700
9.96	924-88-9	Diisopropyl succinate	-	12.94	823-82-5	2,5-Furandicarbaldehyde	-
12.935	504-15-4	Orcinol	-	13.03	611-13-2	Methyl 2-furoate	2703
13.045	611-13-2	Methyl 2-furoate	2703	13.93	118-71-8	Maltol	2656
17.52	10551-58-3	(5-Formyl-2-furyl)methyl acetate	-	17.515	10551-58-3	(5-Formyl-2-furyl)methyl acetate	-
18.74	67-47-0	5-hydroxymethylfurfural	-	18.695	67-47-0	5-hydroxymethylfurfural	-
19.93	118-71-8	Maltol	2656	20.605	4282-34-2	2,5-Thiophenedicarboxylic acid dimethyl este	-
20.945	102-76-1	Triacetin	2007	25.52	498-07-7	Levoglucosan	-
25.52	498-07-7	Levoglucosan	-	29.52	84-69-5	Diisobutyl phthalate	-
29.525	84-69-5	Diisobutyl phthalate	-				
**Sample**	**9**			**Sample**	**10**		
**Retention time/min**	**CAS**	**Compound**	**FEMA**	**Retention time/min**	**CAS**	**Compound**	**FEMA**
3.54	64-18-6	Formic Acid	2487	4.01	64-18-6	Formic Acid	2487
3.795	64-19-7	acetic acid	2006	4.205	64-19-7	acetic acid	2006
4.06	116-09-6	acetol	4462	4.46	116-09-6	acetol	4462
4.3	79-09-4	Propionic acid	2924	4.635	56-81-5	glycerol	2525
4.635	56-81-5	glycerol	2525	5.18	4254-15-3	(*S*)-(+)-1,2-Propanediol	-
4.935	4254-15-3	(*S*)-(+)-1,2-Propanediol	-	5.795	498-60-2	3-Furaldehyde	3737
5.79	498-60-2	3-Furaldehyde	3737	6.92	4412-91-3	3-Furancarbinol	-
6.74	4412-91-3	3-Furancarbinol	-	8.095	497-23-4	2(5*H*)-Furanone	4138
7.78	1192-62-7	2-Acetylfuran	3163	8.765	591-12-8	a-Angelic lactone	3293
8.04	497-23-4	2(5*H*)-Furanone	4138	8.895	16867-04-2	hydroxypyridone	-
9.015	16867-04-2	hydroxypyridone	-	8.99	22913-26-4	Methyl thiophene-3-carboxylate	-
9.28	620-02-0	5-Methyl furfural	2702	9.28	620-02-0	5-Methyl furfural	2702
10.04	924-88-9	Diisopropyl succinate	-	10.025	925-15-5	Dipropyl succinate	-
11.4	80-71-7	Methyl cyclopentenolone	2700	11.95	110-13-4	Acetonylacetone	-
12.96	504-15-4	Orcinol	-	12.765	3658-77-3	4-Hydroxy-2,5-dimethylfuran-3(2*H*)-one	3174
13.09	611-13-2	Methyl 2-furoate	2703	12.945	504-15-4	Orcinol	-
16.63	4940-11-8	Ethyl maltol	3487	13.07	611-13-2	Methyl 2-furoate	2703
17.55	10551-58-3	(5-Formyl-2-furyl)methyl acetate	-	13.885	98-00-0	Furfuryl alcohol	2491
19.1	67-47-0	5-hydroxymethylfurfural	-	17.54	10551-58-3	(5-Formyl-2-furyl)methyl acetate	-
19.95	118-71-8	Maltol	2656	19.14	67-47-0	5-hydroxymethylfurfural	-
25.57	498-07-7	Levoglucosan	-	25.715	498-07-7	Levoglucosan	-
29.52	84-69-5	Diisobutyl phthalate	-				

## Data Availability

Data will be made available on request.

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
