# Peer review of "Application of Gas-Liquid Microextraction (GLME)/GC-MS for Flavour and Fragrance in Ice Cream Detection and Composition Analysis"

_molecules, 2023, doi:10.3390/molecules28020522_

Round 1

Reviewer 1 Report

In terms of originality and scientific novelty, I cannot rate this article highly. The authors used scientific methods known in chemical analysis. This is gas-liquid microextraction (GLME) and gas chromatography with a mass detector GC_MS). The object of research, i.e. edible ice cream obtained on the Chinese market, and the conditions for analyzing the volatile components of this ice cream can be considered as a novelty. The authors optimized the method of preparing ice cream samples for analysis and the basic parameters of chromatographic analyses. I can consider the undertaken problem as original because the chromatographic analysis of volatile compounds in food; in this case in edible ice cream is very important. The obtained results are primarily of practical importance for ice cream producers and consumers. In this area, the results obtained provide advancement in current knowledge.

The research results obtained in the work are correctly interpreted, although they are of a qualitative nature only. The work would be much more valuable if the results were quantitative. The presentation of the results and the content of the article have many shortcomings and need improvement. This includes frequent missing spaces between words, misspellings, etc. The drawings are correct, but I would like to see more chromatograms. The authors included only one chromatogram, and without the description of individual peaks.

The described studies are well-designed and properly performed in technical terms. It is valuable to use Principal Component Analysis and Design Experts 12. Table 2 is interesting and valuable, but its two-column arrangement makes it difficult for the reader to analyze the presented results. In my opinion, the table should be graphically rewritten so that the reader could unambiguously read the results placed in it. Thematically, the article meets the criteria of the scientific journal "Molecules". The results described in this article will certainly be of interest to researchers, scientists, and practitioners dealing with the study of food flavors and fragrances, in particular ice cream.

The weakness of the article is the conclusions, which in my opinion are a summary of the article. They do not contain elements of inference and generalization based on the obtained research results. Conclusions should be redrafted and supplemented. It would be very good if the authors carried out a toxicological analysis of the ingredients detected in the flavors and fragrances of the tested ice cream.
The English language of the article is generally understood but requires significant corrections. Some elements of these corrections have been marked in the attached text of the article and in the comments.
References are written with mistakes and need to be carefully checked and corrected. 

Reviewer 2 Report

This paper reported a gas-liquid microextraction (GLME)/GC-MS method for flavor and fragrance in ice cream detection and composition analysis. The topic of the manuscript is interesting, but there are some deficiencies that need to be modified.

1. The topic lacks novelty and the authors need to explain why this approach is so much better or what if offers that existing similar assays do not.

2. For Ice cream, a kind of frozen dessert, the extraction temperature of 270 oC is not suitable. Some chemical reaction may occur at high temperature.

3. The result of response surface analysis did not analyze in depth.

4. There are issues with the English (syntax, grammar, etc.) throughout the manuscript.

Reviewer 3 Report

Manuscript title: Application of gas-liquid microextraction (GLME)/GC-MS for flavor and fragrance in ice cream detection and composition analysis

The GLME operating conditions (extraction temperature, carrier gas speed, extraction time, condensation temperature) were optimized by single factor analysis and response surface design. Volatile aroma components of ice cream samples were analyzed and compared. However, there are some weaknesses that need to be addressed.

Specific comments:

Most of the format of the spaces and punctuation in the manuscript was not correct, such as Line 53, Line 55, Line 66, etc.

Introduction

1. Author should add the previously published examples on the usage of gas-liquid microextraction in the second paragraph of the introduction section.

2. Author should add the novelty and main content of this work in the last paragraph of the introduction section.

Materials and Methods

Line 58: Suggest to change “Ddifferent” to “different”.

Line 71: What does the word “SHI-MADZU” mean?

In section 2.6: What is the reason for choosing the GLME conditions (extraction temperature of 270 ℃, carrier gas speed of 2.5 mL/min, condensation temperature of -2 ℃, extraction time of 7 min)? The optimal value in single factor analysis and response surface design was also the same as the first design. Is it necessary to do the relevant optimization analysis?

Results and Discussion

Line 109: Suggest to change “With” to “with”.

Line 158: change the font size of “this thesis”.

Table 2:

(1) Please use a footnote to identify “FEMA”.

(2) What is the qualitative identification method for all the volatile compounds obtained by GC-MS?

Line 225: What does the word “melad reaction” mean?

Line 233: suggesting to delete “release a fruity aroma”.

Line 239: suggesting to add a space between “aroma” and “which”.

Section 3.3:

(1) The method of calculating the concentration of samples should be moved to the Materials and Methods.

(2) Please add the explanation of “m” in the formula (1-3).

(3) What is the quantitative result of each designated component? Please add the relevant data.

Round 2

Reviewer 1 Report

After taking into account and introducing the corrections and additions recommended by me, the value of this scientific article has increased significantly. Many things became clear. The manuscript has been significantly revised. After reading through the revised version of the article, I still notice some minor editorial and linguistic shortcomings. For example: no spaces between numbers and units of measurement, typos, etc. I understand that in the editorial process these minor shortcomings will be noticed and corrected. Finally, I believe that due to the extensive research, very good documentation of the results obtained and the scientific and application value of the described research, the article deserves to be published in the scientific journal Molecules.

Author Response

Re: No.  molecules-2067562

Dear reviewer,

  We thank the reviewer for their positive comments and helpful suggestions. We have carefully addressed all concerns raised by the reviewer. We have modified the some minor editorial and linguistic shortcomings.

We are glad that referee found the paper of interest and thanks for excellent comments. We hope that the revised manuscript is acceptable for publication in Molecules.

Reviewer 2 Report

Accept in present form

Author Response

 We thank the reviewer for their positive comments and helpful suggestions. 

Reviewer 3 Report

Results and Discussion

(1) We also don’t know the qualitative identification method for all the volatile compounds obtained by GC-MS in Table 2. In section 3.2.1 and section 3.2.2, the discussions were about the volatiles obtained from the ten samples, it is better to add a summary table of the volatile compounds in 10 samples, not just only two samples (Table 2).

(2) In Section 3.3: Are the data in Figure 6 from one selected ice cream? What is the result of other samples? What is the point of using data from one sample?
